# Dynamic Balance Control of Double Gyros Unicycle Robot Based on Sliding Mode Controller

**DOI:** 10.3390/s23031064

**Published:** 2023-01-17

**Authors:** Yang Zhang, Hongzhe Jin, Jie Zhao

**Affiliations:** State Key Laboratory and System, Harbin Institute of Technology, Harbin 150001, China

**Keywords:** unicycle robot, sliding mode control, dynamics, mobile robot

## Abstract

This paper presents a doublegyroscope unicycle robot, which is dynamically balanced by sliding mode controller and PD controller based on its dynamics. This double−gyroscope robot uses the precession effect of the double gyro system to achieve its lateral balance. The two gyroscopes are at the same speed and in reverse direction so as to ensure that the precession torque of the gyroscopes does not interfere with the longitudinal direction of the unicycle robot. The lateral controller of the unicycle robot is a sliding mode controller. It not only maintains the balance ability of the unicycle robot, but also improves its robustness. The longitudinal controller of the unicycle robot is a PD controller, and its input variables are pitch angle and pitch angular velocity. In order to track the set speed, the speed of the unicycle robot is brought into the longitudinal controller to facilitate the speed control. The dynamic balance of the designed double gyro unicycle robot is verified by simulation and experiment results. At the same time, the anti−interference ability of the designed controller is verified by interference simulation and experiment.

## 1. Introduction

As a wheeled mobile robot, the unicycle robot has high flexibility due to its contact with the ground being a single point of contact. However, the characteristics of the unicycle robot, such as difficult balance control, high coupling, weak anti−interference ability, also bring great challenges to the research into unicycle robots. There are many researchers studying unicycle robots. According to the classification of lateral balance of unicycle robots, there are the following: Stanford and MIT use horizontal rotors to achieve the balance control of the unicycle robot [1,2]; The gyrover unicycle robot designed by Carnegie Mellon University and Yangsheng Xu team controls the robot’s lateral balance according to the high−speed flywheel [3,4,5,6]; Pusan National University of Korea uses vertical rotor structure to achieve the balance of the unicycle robot [7]; Chiba University of Japan [8] and the Asian Institute of Technology in Thailand [9] use the precession effect of double gyros to achieve the balance control of the unicycle robot; J. Shen and D. Hong applied universal wheel to realize the balance control of the designed unicycle robot [10,11]. Many studies on the unicycle robot focus on its static balance control. However, as a mobile robot, its mobility and anti−interference ability are also worth studying. Kwok Wai Au used the state feedback controller to realize the displacement tracking of the Grover unicycle robot [12]. Umashankar Nagarajan et al. used PID controller to realize the dynamic balance control of the designed unicycle robot through off−line motion trajectory [13]. It can be seen from previous studies that the research on dynamic balance control of unicycle robot is also important. Due to the point contact movement of the unicycle robot, its anti−interference ability is worse than other mobile robots. However, many previous studies on the controller of the unicycle robot focus on the balance control of the unicycle robot, while neglecting to enhance its anti−interference ability. Therefore, the dynamic anti−interference balance control of the unicycle robot is studied in this paper.

As a highly complex structure of the unicycle robot, the double gyros unicycle robot has the characteristics of control difficulty, serious self vibration and interference. Compared with other unicycle robots, the double gyros has better balance characteristics due to its gyro effect. Therefore, the dynamic balance control and anti−interference of the designed unicycle robot are studied in this paper. The dynamics of the double gyros unicycle robot has been completed in previous research [14]. In this paper, according to the dynamic model of the unicycle robot, sliding mode controller is used to control its lateral dynamic balance. The sliding mode controller has anti−interference ability to ensure that the unicycle robot can still maintain balance when facing lateral interference under the dynamic motion. Its dynamic anti−interference ability has been verified in simulation and experiment. Longitudinal balance takes pitch angle, pitch angular speed and the speed of unicycle robot as controller inputs. The PD controller is used to ensure the dynamic speed tracking and longitudinal balance of the unicycle robot. According to the corresponding relationship between the pitch angle and the bottom wheel speed in the longitudinal dynamics, at the target speed, the corresponding target pitch angle is given to ensure balance.

The rest of this paper is as follows. Section 2 is the model of the designed unicycle robot, which includes lateral dynamics model, longitudinal dynamics model and experimental platform model. Section 3 is the controller designed according to the dynamics of the unicycle robot. Section 4 provides the simulation results. Section 5 provides the experimental results. Section 6 presents the conclusion and future work.

## 2. Model

In this section, based on the Lagrangian dynamics method, the dynamic model of the designed double gyros unicycle robot is decomposed into two parts: lateral dynamics and longitudinal dynamics. The dynamics of the double gyros unicycle robot are shown in previous work [14]. Its Lagrangian dynamics are as follows:(1)$$M\left(q\right)\ddot{q}+N\left(\dot{q},q\right)=Q.$$

Mq∈ℝ6×6 and Nq˙,q∈ℝ6×1 the inertia matrix and nonlinear term, respectively, where
Mq=m11m12m13m140m16m21m22m2300m26m31m32m33m3400m410m43m44000000m550m61m62000m66
Nq˙,q=n1 n2 n3 n4 n5 n6T.

This paper only considers the linear dynamics of the unicycle robot, so the influence of yaw angle on the unicycle robot is ignored. For the system, ***q*** and Q are
$$q={\left(\varphi ,\delta ,\theta ,\omega ,\alpha ,\beta \right)}^{T},\;Q={\left(0,0,0,\tau_{\omega },\tau_{\alpha },\tau_{\beta }\right)}^{T}$$
where $\tau_{\omega },\;\tau_{\alpha }$ and $\tau_{\beta }$ are torque of the bottom, torque of precession system and torque of gyro system, respectively. Next, the dynamics are decomposed into lateral dynamics and longitudinal dynamics.

### 2.1. Lateral Dynamics

Figure 1 shows the lateral model of the unicycle robot. It consists of double gyro rotation and precession system. The two gyroscopes have the same speed and opposite direction. The precession angular speed of these two gyroscopes is the same, and the direction is opposite. This can ensure that the gyroscopic moment generated by the precession of the gyroscope can offset the longitudinal interference. In the figure, δ Is the roll angle of the unicycle robot, $\alpha_{1},\;\alpha_{2}$ are the precession angles of the left and right sides of the unicycle robot. $\beta_{1},\beta_{2}$ are the rotation angles of the left and right gyroscopes, respectively.

According to the dynamic Equation (1), the gyro rotation is assumed to be constant, and the influence of yaw angle on the side direction of the unicycle robot is ignored. The lateral dynamic is as follows:(2)$$m_{22}\ddot{\delta }+m_{23}\ddot{\theta }+n_{2}=0$$

When the unicycle robot is dynamically balanced, its lateral direction is regarded as an inverted pendulum model, so the influence of pitch angle and bottom wheel speed can be ignored. Where parameter $I_{1z}-I_{1x}$ is small, so the lateral dynamic equation can be solved as follows:(3)$$m_{22}\ddot{\delta }+2\dot{\beta }\dot{\alpha }c\alpha I_{1z}-G_{roll}\delta =0$$

In order to simplify the dynamic equations, cx and sx mean $cos\left(x\right)$ and $sin\left(x\right)$*,* respectively. Where $I_{1z}$ is the moment of inertia of gyroscopes’ center of gravity to *Z* axe for left, $G_{roll}$ is the lateral gravity component, and its equation is as follows:(4)$$G_{roll}=M_{w}gR_{W}+M_{b}g\left(L_{b}+R_{W}\right)+2M_{p1}g\left(L+R_{W}\right)+2M_{1}g\left(L+R_{W}\right)$$
where $M_{w},\;M_{b},\;M_{p1}\;\mathrm{and}\;M_{1}$ are mass of the wheel, mass of the frame, mass of the precession frame and mass of the gyroscope, respectively. $R_{W},\;L_{b}\;\mathrm{and}$ L are radius of the wheel, distance of frame’s center of gravity from center of wheel and distance of gyro’s center of gravity from precession frame’s center of gravity for left. g is the unit of gravity. It can be seen from the lateral dynamic equation that the roll angle is mainly affected by the rotation speed and precession angular speed of the double gyros. If precession angle α is too large, the lateral precession torque will be smaller. Therefore, in the lateral controller, the precession angular velocity should be used to control the lateral balance of the unicycle robot when the gyros’ rotation speed is constant. At the same time, in order to keep the precession angle as small as possible to ensure lateral controllability, the precession angle is regarded as the input variable in the lateral controller.

### 2.2. Longitudinal Dynamics

Figure 2 shows the longitudinal model of the unicycle robot. It is composed of the bottom wheel and frame. In the figure, θ is the pitch angle of the unicycle robot and ω is the rotation angle of the bottom wheel of the unicycle robot.

According to the dynamic Equation (1), ignore the influence of yaw angle on the longitudinal direction of the unicycle robot. The longitudinal dynamic is as follows:(5)$$m_{32}\ddot{\delta }+m_{33}\ddot{\theta }+m_{34}\ddot{\omega }+n_{3}=0$$

When the unicycle robot is dynamically balanced, its longitudinal direction is regarded as an inverted pendulum model, so the influence of roll angle can be ignored. Where parameter $I_{1z}-I_{1x}$ is small, so the longitudinal dynamic equation can be solved as follows:(6)$$m_{33}\ddot{\theta }+m_{34}\ddot{\omega }-G_{pitch}\theta =0$$
where $G_{pitch}$ is the longitudinal gravity component, and its equation is as follows:(7)$$G_{pitch}=M_{b}gL_{b}+2M_{p1}gL+2M_{1}gL$$

It can be seen from the longitudinal dynamic equation that its longitudinal balance is mainly affected by the bottom wheel. Although the precession angle rotation of the double gyros also affects the longitudinal direction, $\left(4s\alpha c\alpha I_{1x}-4s\alpha c\alpha I_{1z}\right)$ is small, and its impact can be ignored. Therefore, in the longitudinal control, the torque of the bottom wheel is taken as the longitudinal control variable, and in order to ensure that the bottom wheel can track the set speed, the speed of the unicycle is added to the controller.

### 2.3. Experimental Platform Model

As shown in Figure 3, it is the CAD model of the designed unicycle robot. It is mainly composed of three parts: gyro system, bottom wheel system and frame system. In the gyro system, the gyro is driven by a brushless DC motor, and precession rotation is achieved by gear transmission. The left and right gyroscopes are the same. In the frame system, the symmetrical structure is used to distribute the drive motor to ensure the balance of the unicycle robot, and the belt drive is used to drive the gear to realize the precession rotation. The left and right precession rotation is driven by the same gear input shaft to ensure the same precession speed. Moreover, its driving motor is a DC servo motor on the frame. The bottom wheel is also driven by the DC servo motor on the frame, and its transmission mode is belt. The left and right sides of the bottom pulley belt drive have the same structure to ensure the symmetry of the unicycle robot in the structure. In the experiment section, the experiments are completed according to the designed unicycle robot platform.

## 3. Design of the Controllers

In this section, the lateral and longitudinal controllers are designed according to the dynamic model of the double gyros unicycle robot. The control block diagram of the unicycle robot is shown in Figure 4. According to the dynamic model, the lateral controller is a sliding mode controller to ensure the dynamic balance of the unicycle robot and increase its anti−interference ability. The longitudinal controller is a PD controller, in which a speed tracker is added to ensure the dynamic speed tracking of the unicycle robot. The sliding mode controller has excellent anti−interference ability. In order to enhance the anti−interference ability of the unicycle robot, the lateral controller is designed as a sliding mode controller. At the same time, it helps to improve the lateral balance ability. The longitudinal controller is used to ensure the longitudinal balance of the unicycle robot while tracking the set speed. The longitudinal control state variable has not only pitch angle, but also speed value. In order to simplify the lateral controller and let the longitudinal balance track the set speed at the same time, the longitudinal controller is designed as a PD controller.

### 3.1. Lateral Controller

The lateral dynamic equation is shown in Equation (3). In the dynamic controller of the unicycle robot, the rotation angle speed of the gyroscopes is constant ($\ddot{\beta }=0$). The variable $\dot{\beta }$ in dynamics can be regarded as a constant value. Therefore, the lateral dynamic equation can be changed as follows:(8)$$\ddot{\delta }=a_{\delta }\delta -b_{\delta }\dot{\alpha }c\alpha$$
where $a_{\delta }$ and $b_{\delta }$ are lateral dynamic parameters, both of which are constant values. The roll angle error equation is as follows:(9)$$e_{\delta }=\delta -\delta_{d}$$
where $\delta_{d}$ is the desired roll angle. According to the roll angle error value $e_{\delta }$, its lateral sliding surface is as follows:(10)$$s_{\delta }={\dot{e}}_{\delta }+\lambda e_{\delta }$$
where λ is a constant parameter. The derivative of the lateral sliding surface can be obtained as follows:(11)$${\dot{s}}_{\delta }={\ddot{e}}_{\delta }+\lambda {\dot{e}}_{\delta }$$

Bring Equation (8) into Equation (11) and take precession angular velocity $\dot{\alpha }$ as the output value of lateral balance controller. For lateral balance, if ${\dot{s}}_{\delta }\to 0$, the expected input value $\hat{\dot{\alpha }}$ is as follows:(12)$$\hat{\dot{\alpha }}=\left(a_{\delta }\delta +\lambda {\dot{e}}_{\delta }-{\ddot{\delta }}_{d}\right)/b_{\delta }c\alpha$$

Since the parameter values $a_{\delta }$ and $b_{\delta }$ are estimated values, they are not accurate. Due to the error between the actual precession angular velocity $\dot{\alpha }$ and expected precession angular velocity $\hat{\dot{\alpha }}$, in order to reduce the interference caused by the error to the unicycle robot and increase the anti−interference of the controller, a discontinuous term is added between the actual and expected precession angular velocity to make it swing in the designed sliding surface $s_{\delta }$. The precession angular velocity can be obtained as follows:(13)$$\dot{\alpha }=\hat{\dot{\alpha }}-k_{\delta }sgn\left(s_{\delta }\right)$$

$k_{\delta }$ is the coefficient of discontinuous function $sgn\left(s_{\delta }\right)$. Where function $sgn\left(s_{\delta }\right)$ is a discontinuous function, the equation is as follows:(14)$$sgn\left(s_{\delta }\right)=\left\{\begin{array}{c} +1\;\;\;\;\;s_{\delta }>0\\ -1\;\;\;\;\;s_{\delta }<0 \end{array}\right.$$

Thus, a lateral sliding mode controller can be obtained.

### 3.2. Longitudinal Controller

The longitudinal dynamic equation is shown in Equation (6). It can be seen from the dynamic equation that the longitudinal balance is affected by the bottom wheel. If the longitudinal direction of the unicycle robot is regarded as the inverted pendulum model, the torque of the bottom wheel is as follows:(15)$$\tau_{\omega }=m_{\theta }\ddot{\omega }$$

Where $m_{\theta }$ is the longitudinal component of the mass of the unicycle robot. Therefore, the bottom wheel torque $\tau_{\omega }$ can be regarded as the output value of the longitudinal controller. The longitudinal balance controller is a PD controller, and its control equation is as follows:(16)$$\tau_{\omega }=\kappa_{P}\theta +\kappa_{D}\dot{\theta }$$

$\kappa_{P}$ and $\kappa_{D}$ are the pitch angle and pitch angle velocity coefficients in the longitudinal PD controller, both of which are constant. In order to track the set bottom wheel speed, the bottom wheel speed tracking is added on the basis of PD controller. The control equation can be obtained as follows:(17)$$\tau_{\omega }=\kappa_{P}\theta +\kappa_{D}\dot{\theta }+\rho_{P}{\dot{e}}_{\omega }$$
where $\rho_{P}$ is the constant coefficient of bottom wheel speed error and ${\dot{e}}_{\omega }$ is the bottom wheel speed error, and its equation is as follows:(18)$${\dot{e}}_{\omega }=\dot{\omega }-{\dot{\omega }}_{d}$$

$\omega_{d}$ is the desired bottom wheel speed. The unicycle robot does not move at a constant speed in the process of speed tracking. When the bottom wheel has acceleration, the pitch angle must be disturbed. In this case, if the target pitch angle set by the longitudinal PD control is 0, it is bound to cause certain interference to the pitch angle. In the longitudinal balance controller, the corresponding equation is designed for the target forward speed value and the target pitch angle value. The equation is as follows:(19)$$\theta_{d}=\mu_{\omega }{\dot{\omega }}_{d}$$

In order to control the speed of the unicycle robot, the angular velocity of the bottom wheel is converted into the speed of the robot. Since the relationship between the robot speed and the bottom wheel angular velocity is $v=R_{w}\dot{\omega }$, Equation (19) can be transformed as follows:(20)$$\theta_{d}=\mu v_{d}$$

$v_{d}$ is the target speed of the unicycle robot. μ Is the constant coefficient of Equation (20). Therefore, the longitudinal controller changes as follows:(21)$$\tau_{\omega }=\kappa_{P}e_{\theta }+\kappa_{D}{\dot{e}}_{\theta }+\rho e_{v}$$
where $e_{v}$ is the error of the speed of the unicycle robot. It is as follows:(22)$$e_{v}=v-v_{d}$$

The error value of pitch angle is as follows:(23)$$e_{\theta }=\theta -\theta_{d}$$

## 4. Simulation

In order to test the balance ability and anti−interference ability of the designed controller on the double gyros unicycle robot, it is verified in the three−dimensional simulation environment. The simulation software is Vrep. The designed CAD model is brought into the simulation software, and the connection pair and model parameters are set to ensure the authenticity of the simulation. The rotation speeds of the double gyros are set to 7000 rpm and the direction is opposite. The precession angular velocities on the left and right sides are the same and in opposite directions. The precession angular velocity is taken as the lateral balance control output, and the bottom wheel torque is taken as the longitudinal balance control output. The set tracking speed of the bottom wheel is 0.08 m/s. In the simulation, a 3N pulse interference is generated on the side of the unicycle robot at 10 s, and the anti−interference ability of the designed controller is tested. The initial coefficients of the controller are given according to the estimated parameters in the dynamics of the designed unicycle robot, and then the performance of the unicycle robot is achieved through minor adjustment. The sliding mode controller can enhance the balance ability and anti−interference ability of the unicycle robot by adjusting the coefficient of discontinuous function $k_{\delta }$. The simulation curves are shown in Figure 5. Figure 5a shows the roll angle in the simulation. It can be seen that the roll angle of the unicycle robot fluctuates slightly and keeps balance. After interference, the roll angle has a large swing and immediately returns to a stable state. Figure 5b shows the pitch angle in the simulation. Due to the unicycle robot displaying a dynamic movement in the process of moving forward, it has a certain value in the pitch angle to ensure the longitudinal balance of the unicycle robot. After the lateral interference has a small interference to the longitudinal, the pitch angle returns to the equilibrium state immediately. Figure 5c shows the precession angle in the simulation. After the lateral interference, the precession angle returns to the balance state immediately after the large swing to maintain the dynamic balance of the unicycle robot. Figure 5d shows the speed curve of the unicycle robot in the simulation. It can be seen that the unicycle robot can track the set moving speed of 0.08 m/s with minimal gap. According to the simulation, it can be seen that the designed controller has the ability to maintain balance with large disturbances, and the unicycle robot can also track the set speed.

In the simulation, continuous stochastic interference is added for comparison. After the unicycle robot is balanced for 4 s, a stochastic pulse interference within 1–2N is added every 2 s. PD controller is added as comparison controller to the simulation. The longitudinal controller of PD controller is the same as the designed controller. The lateral controller is a PD controller composed of roll angle and roll angular velocity. However, in the dynamic balance simulation of the unicycle robot under 3N pulse disturbance, PD controller is difficult to keep the balance. Therefore, the 3N pulse interference curve only has the designed controller curves. The curves are shown in Figure 6. Figure 6a–d are the comparison curves of roll angle, pitch angle, precession angle and speed, respectively. In the case of continuous stochastic interference, the pitch angle and speed of PD controller and designed controller have little influence. In the roll angle, the designed controller has less vibration and is more stable than the PD controller in the continuous stochastic interference. In the precession angle, the designed controller has smaller amplitude and faster in convergence speed than PD controller. It can be seen that the designed controller has better anti−interference ability and is more stable than the PD controller in the case of dynamic balance of the unicycle robot in the continuous stochastic interference.

## 5. Experimental

The dynamic balance ability and stability of the controller to the robot are verified by using the designed experiment platform of the double gyros unicycle robot. The roll angle and pitch angle of the unicycle robot are obtained by the gyroscope sensor mpu9250 installed in the center of the unicycle robot. The precession angle and the angle of the bottom wheel are calculated according to the encoder on the precession drive motor and bottom wheel drive motor, respectively. The gyroscopes have constant speed and the same left, right and opposite direction. The experiment mainly includes dynamic balance experiment and dynamic interference experiment. The dynamic interference experiment is to verify the dynamic anti−interference ability with a 0.18 kg mass block by placing it in the lateral, middle and longitudinal directions when the unicycle robot is moving forward dynamically.

Figure 7 shows the dynamic balance experiment curve of the unicycle robot. Figure 7a–d show the roll angle curve, the pitch angle curve, the precession angle curve and the speed curve of the unicycle robot in the dynamic balance experiment, respectively. It can be seen from the figures that with the designed controller, the roll angle of the double gyros unicycle robot is stable near the zero position and fluctuates slightly. As the longitudinal dynamic balance controller is designed, the pitch angle has a corresponding tilt angle while the speed is tracked. In the experiment, the set speed is 0.2 m/s. According to Equation (18), the corresponding pitch angle is shown in Figure 7b. Although the speed fluctuates greatly due to the influence of the bottom wheel transmission mode, the speed can basically track the set speed value. The PD controller also can keep the unicycle robot balanced and stable.

Figure 8 shows the curves of the left side interference dynamic experiment of the unicycle robot. Place the 0.18 kg mass on the left side of the unicycle robot (90 cm from the centre) when the unicycle robot is balanced for 10 s. In the dynamic balance experiment with 0.18 kg mass block placed, the PD controller is difficult to keep the unicycle robot balanced, so the step interference dynamic balance experiment of the PD controller is not shown here. The unicycle robot is affected by the step interference caused by small gravel falling on it. The mass block is placed at the unicycle robot when it is moving forward dynamically to simulate the anti−interference ability of the unicycle robot in the face of step interference. The 0.18 kg mass block is used to simulate the balance ability of the unicycle robot under the influence of large step interference. Figure 8a–d show the roll angle curve, the pitch angle curve, the precession angle curve and the speed curve of the dynamic interference balance of the unicycle robot, respectively. It can be seen that the roll angle and precession angle are greatly affected by interference. After interference, the roll angle keeps balance after fluctuation, and the precession angle keeps balance after large swing. The lateral interference has little effect on the pitch angle, and the speed has slight interference, but the speed remains stable after balance.

Figure 9 shows the curves of the middle interference dynamic experiment of the unicycle robot. Place the 0.18 kg mass on the middle of the unicycle robot when the unicycle robot is balanced for 10 s. Figure 9a–d show the roll angle curve, the pitch angle curve, the precession angle curve and the speed curve of the dynamic interference balance of the unicycle robot, respectively. It can be seen from the curves that when the mass block is placed in the middle of the unicycle robot, the longitudinal and lateral effects are very small.

Figure 10 shows the curves of the interference dynamic experiment on the rear of the unicycle robot. Place the 0.18 kg mass on the rear of the unicycle robot (43 cm from the centre) when the unicycle robot is balanced for 12 s. Figure 10a–d show the roll angle curve, the pitch angle curve, the precession angle curve and the speed curve of the dynamic interference balance of the unicycle robot, respectively. It can be seen from the curves that the interference on the rear has little influence on the lateral of the unicycle robot, and the roll angle and precession angle fluctuate slightly. Rear interference has great influence on the longitudinal direction. After the interference, the pitch angle becomes larger and has a large swing. After the interference, the pitch angle has a large swing and is balanced at a larger angle value. The speed is greatly increased after interference, but it is still within the set speed range.

The dynamic balance experiment curves with continuous stochastic interference are shown in Figure 11. Figure 11a–d are the comparison curves of roll angle, pitch angle, precession angle and speed, respectively. It can be seen from the curves that continuous stochastic interference has a greater impact on roll angle and precession angle. However, after the interference, the roll angle and precession angle can still quickly recover to balance. Compared with PD controller, the designed controller has smaller amplitude in roll angle and precession angle. The interference also affects the pitch angle and speed. The pitching angle can still recover the balance state after interference. Although there is a large deviation in speed after interference, it can slowly recover to the tracking speed. Compared with PD controller, the designed controller has smaller oscillation and faster recovery when the pitch angle is subject to continuous stochastic interference and the speed fluctuates less.

According to the dynamic test and interference experiment of the designed controller on the double gyros unicycle robot, it can be seen that the designed controller can track the set speed of the unicycle robot and has the ability of anti−interference. The interference on the rear has a strong interference with the longitudinal direction of the unicycle robot, but the robot can keep balance and track the set speed after the interference. It can be seen that the double gyros unicycle robot can maintain dynamic balance, have anti−interference ability and track the set speed by using the designed controller.

## 6. Conclusions

In this paper, the dynamic balance controller is designed according to the dynamic model of the double gyros unicycle robot. The lateral controller is a sliding mode controller, which can improve the anti−interference ability of the unicycle robot. In the longitudinal controller, PD controller is used to balance. At the same time, due to the relationship between the pitch angle and moving speed, the corresponding equation is designed, and the designed speed tracking equation is taken into the longitudinal controller to ensure that the speed can track the designed speed. In the simulation, the dynamic balance and anti−interference ability of the designed controller are verified. According to the dynamic balance experiment, lateral, middle and longitudinal interference experiments, the dynamic balance and anti−interference ability of the designed controller on the double gyros unicycle robot are verified. The contribution of this paper is to design the lateral sliding film controller and the longitudinal speed tracking controller for the double gyros unicycle robot to achieve its dynamic balance ability and dynamic anti−interference ability. In the latter research, we will study the yaw angle control and autonomous motion of the double gyros unicycle robot.

## Figures and Tables

**Figure 1 sensors-23-01064-f001:**
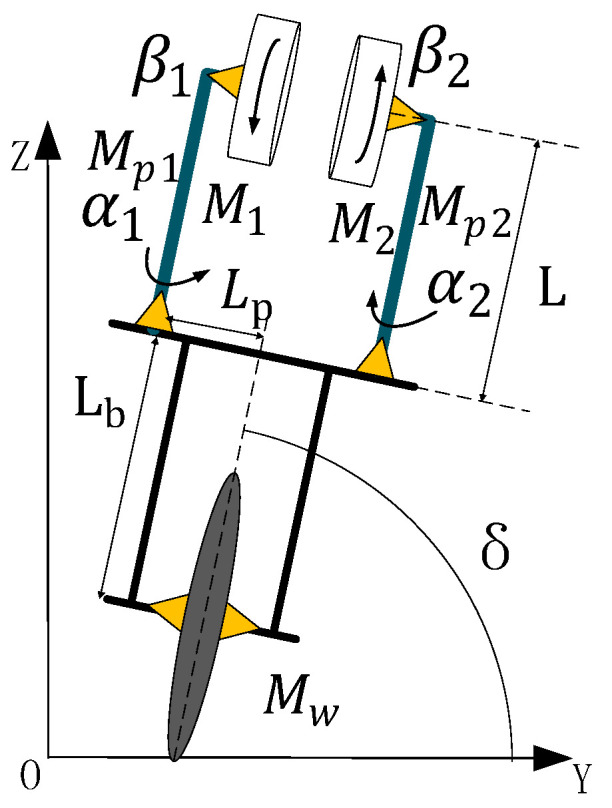
Lateral model of the unicycle robot.

**Figure 2 sensors-23-01064-f002:**
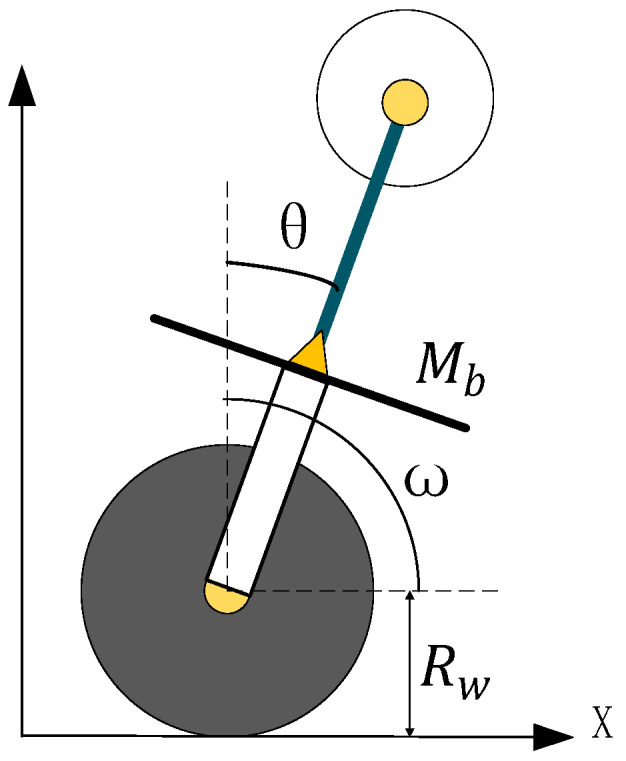
Longitudinal model of the unicycle robot.

**Figure 3 sensors-23-01064-f003:**
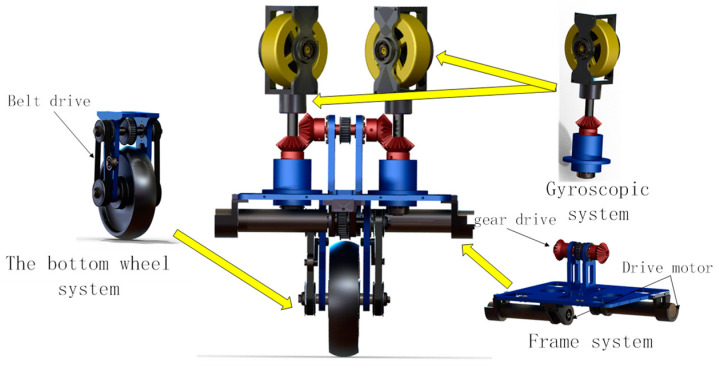
CAD model of the unicycle robot.

**Figure 4 sensors-23-01064-f004:**
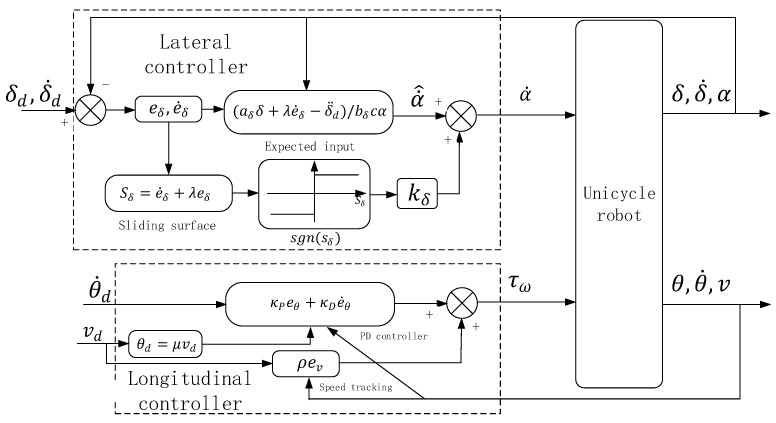
The control block diagram of the unicycle robot.

**Figure 5 sensors-23-01064-f005:**
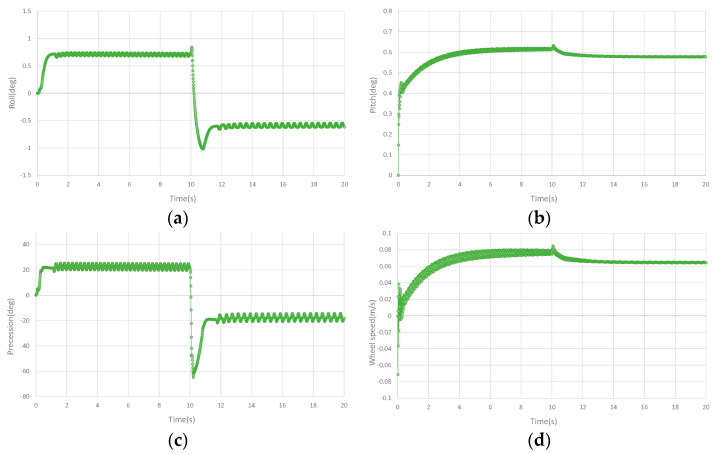
Curves of the unicycle robot in simulation. (**a**) The roll angle curve in the simulation. (**b**) The pitch angle curve in the simulation. (**c**) The precession angle curve in the simulation. (**d**) The speed curve in the simulation.

**Figure 6 sensors-23-01064-f006:**
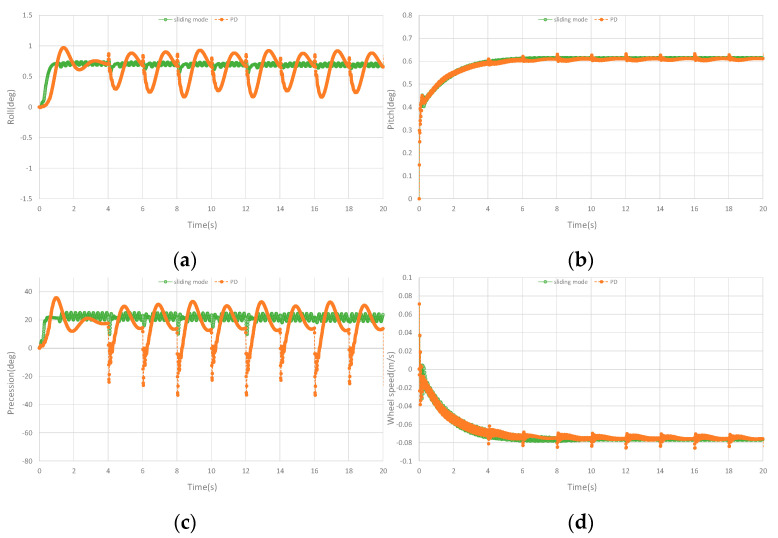
Curves of stochastic pulse interference in simulation. (**a**) The roll angle curve of stochastic pulse interference in the simulation. (**b**) The pitch angle curve of stochastic pulse interference in the simulation. (**c**) The precession angle curve of stochastic pulse interference in the simulation. (**d**) The speed curve of stochastic pulse interference in the simulation.

**Figure 7 sensors-23-01064-f007:**
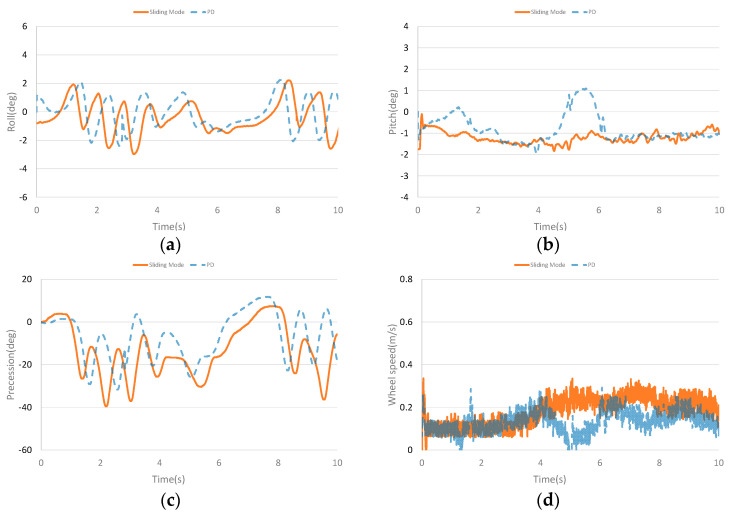
Dynamic balance curves. (**a**) The roll angle curve in dynamic balance experiment. (**b**) The pitch angle curve in dynamic balance experiment. (**c**) The precession angle curve in dynamic balance experiment. (**d**) The speed curve in dynamic balance experiment.

**Figure 8 sensors-23-01064-f008:**
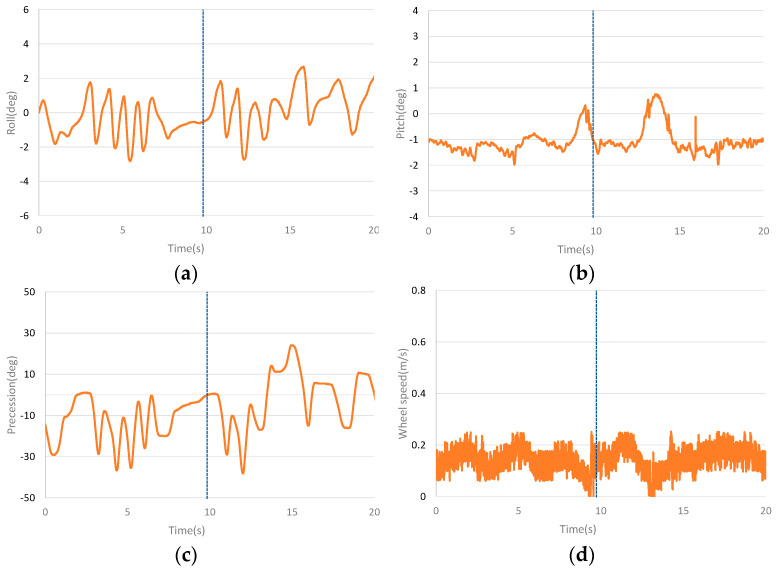
Dynamic balance curves after placing a mass block laterally. (**a**) The roll angle curve of lateral placement of material block experiment. (**b**) The pitch angle curve of lateral placement of material block experiment. (**c**) The precession angle curve of lateral placement of material block experiment. (**d**) The speed curve of lateral placement of material block experiment.

**Figure 9 sensors-23-01064-f009:**
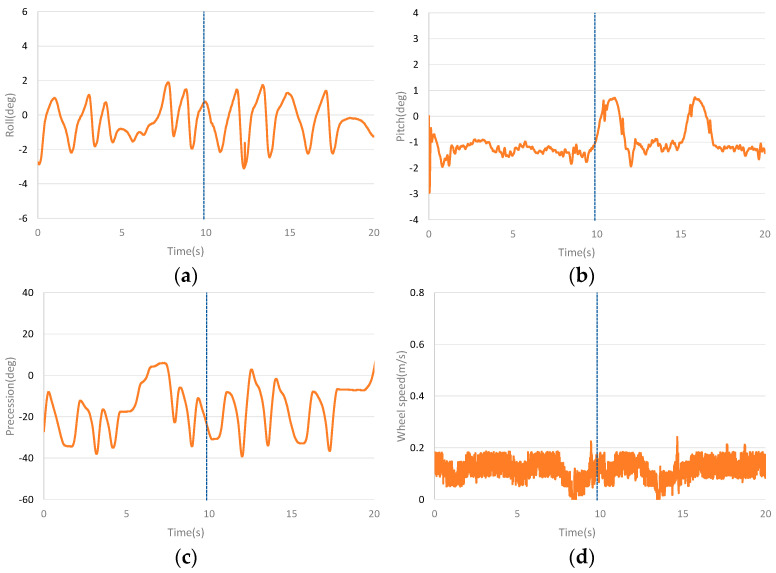
Dynamic balance curves after placing a mass block in middle. (**a**) The roll angle curve of placing material block in the middle. (**b**) The pitch angle curve of placing material block in the middle. (**c**) The precession angle curve of placing material block in the middle. (**d**) The speed curve of placing material block in the middle.

**Figure 10 sensors-23-01064-f010:**
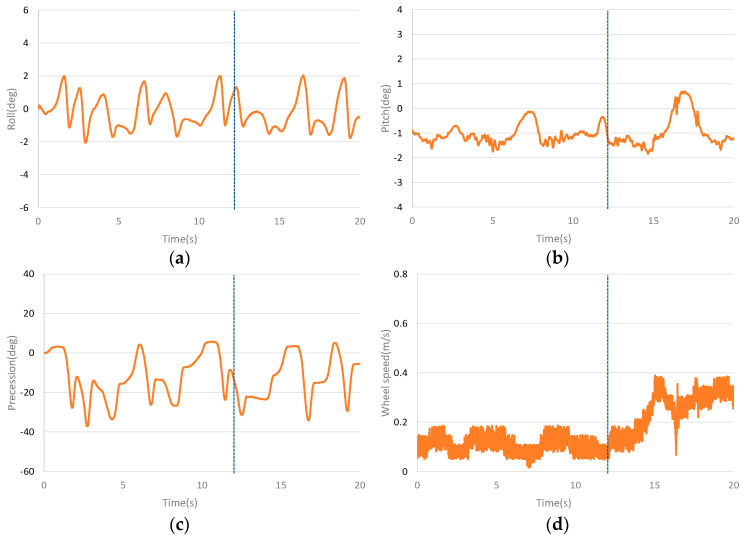
Dynamic balance curves after placing a mass block longitudinally. (**a**) The roll angle curve of placing material block longitudinally. (**b**) The pitch angle curve of placing material block longitudinally. (**c**) The precession angle curve of placing material block longitudinally. (**d**) The speed curve of placing material block longitudinally.

**Figure 11 sensors-23-01064-f011:**
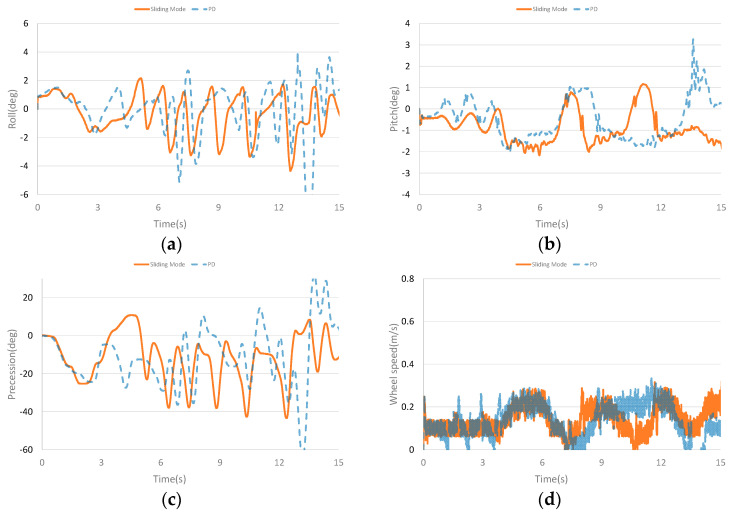
Curves of stochastic pulse interference experiment. (**a**) The roll angle curve of stochastic pulse interference experiment. (**b**) The pitch angle curve of stochastic pulse interference experiment. (**c**) The precession angle curve of stochastic pulse interference experiment. (**d**) The speed curve of stochastic pulse interference experiment.

## Data Availability

The data is unavailable due to privacy.

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
