# Peer review of "Dynamic Balance Control of Double Gyros Unicycle Robot Based on Sliding Mode Controller"

_sensors, 2023, doi:10.3390/s23031064_

Round 1

Reviewer 1 Report

This paper presents a dynamic balance control method based on the sliding mode controller, which has the ability of anti-interference. The introduction states the purpose of the paper, and the relation between the paper and the previous works is clearly explained. The feasibility of the presented method is illustrated via simulation and experiment.

The following are my specific comments:

(1) It is stated in the introduction that “There are many researchers studying unicycle robots”. What are the main limitations of the existing control methods? It is better to provide some remarks.

(2) Some denotations are not defined in equations (1), (3), (4) and (7). The definitions of the denotations should be given when they first appear.

(3) Why are the sliding mode controller and the PD controller designed for lateral controller and longitudinal controller respectively? How is the coefficients of the sliding mode controller and the PD controller designed? It is better to give some explanations.

(4) To illustrate the advantage of the dynamic balance controller based on sliding mode controller, the authors are suggested to evaluate the performance of the presented control method in comparison with the existing method by simulation or experiment.

(5) Why is the interference of the double gyros unicycle robot simulated using a 0.18 kg mass? How is it related to the problem in practical systems? Try to provide some explanations.

(6) Some stochastic values of the control errors obtained from the simulation and experiment should be given to illustrate the high performance of the presented control method.

(7) The main innovations of the paper should be summarized in Conclusion.

(8) There are some typo errors in the paper, such as the sentence after equation (10), the first sentence in Section 5.

Author Response

Responses to Review Remarks

Dear reviewer,

First of all, I would like to express my respect for your responsible and careful peer review. Thank you for your valuable comments and allowing a resubmission of our manuscript.

We are uploading (a) our point-by-point response to the comments (below) , (b) an updated manuscript with Revised.

We have revised the paper in our best efforts. However, we are open to revise further should any further suggestion from the reviewer be offered. Thank you for your time and efforts on this paper.

  1. MAJOR DIFFERENCE FROM THE PREVIOUS VERSION
  2. “Introduction” are revised;
  3. The variables not defined in equations (1), (3), (4) and (7) are interpreted;
  4. The lateral controller is designed as sliding mode controller, and the longitudinal controller is designed as PD controller. The explanation is added in Section 3;
  5. The coefficient design description of sliding mode controller and PD controller is added in Section 4;
  6. In section 4, continuous stochastic interference simulation is added and compared with the traditional PD controller;
  7. The comparison of traditional PD controllers is added to the original Fig. 6;
  8. In section 5, continuous stochastic interference experiment is added and compared with the traditional PD controller;
  9. In the section 5, it is explained that 0.18Kg mass block is selected to simulate external interference;;
  10. The conclusion part explains the innovation of this paper;
  11. Correct the sentence errors in the paper.

  1. RESPONSE TO THE REVIEWER

This paper presents a dynamic balance control method based on the sliding mode controller, which has the ability of anti-interference. The introduction states the purpose of the paper, and the relation between the paper and the previous works is clearly explained. The feasibility of the presented method is illustrated via simulation and experiment.

Reply:

Thank you very much for your efforts on handling this paper and for whole evaluation of our work. We have addressed all of the comments raised, and revised the paper accordingly.

The following are my specific comments:

(1) It is stated in the introduction that “There are many researchers studying unicycle robots”. What are the main limitations of the existing control methods? It is better to provide some remarks.

Reply:

Thank you very much for your comments to make this article more rigorous.

In the introduction, the unicycle robots with different lateral balance modes are introduced first. After that, the research on various controllers of the unicycle robot is described. In these researches on the controller of the unicycle robot, the static and dynamic balance performance is mainly concerned, while the research on enhancing the anti-interference ability of the unicycle robot is less. Due to the point contact movement of the unicycle robot, its anti-interference ability is worse than other mobile robots. However, many previous researches on the controller of the unicycle robot focus on the balance control of the unicycle robot, while neglecting to enhance its anti-interference ability. Therefore, the dynamic anti-interference balance control of the unicycle robot is studied in this paper.

The first paragraph of the introduction of the previous version paid too much attention to the description of the past research on unicycle robots, and weakened the contribution of this manuscript. Therefore, in the first paragraph of the introduction of the revised manuscript, The limitations of the existing research on unicycle robot controller are described.

The modified parts in the introduction have been updated in the Revised Manuscript.

(2) Some denotations are not defined in equations (1), (3), (4) and (7). The definitions of the denotations should be given when they first appear.

Reply:

Thank you for your careful review and valuable suggestion to improve the level of this paper.

First, we would like to apologize for the confusion caused by us. In the revised manuscript, we have described the variables not defined in equations (1), (3), (4) and (7). Change as follows:

  1. In equation (1), q and Q are:

.

Where  and  are torque of the bottom, torque of precession system and torque of gyro system, respectively.

  1. In equation (3), In order to simplify the dynamic equations, and mean  and , respectively. Where  is moment of inertia of gyroscopes’ center of gravity about Z axe for left.
  2. In equation (4), Where are mass of the wheel, mass of the frame, mass of the precession frame and mass of the gyroscope respectively.  are radius of the wheel, distance of frame’s center of gravity from center of wheel and distance of gyro’s center of gravity from precession frame’s center of gravity for left.  is the unit of gravity.
  3. In equation (7), Variables are described in equation (4).

The modified parts have been updated in the Revised Manuscript. Sorry again for the confusion caused by us.

(3) Why are the sliding mode controller and the PD controller designed for lateral controller and longitudinal controller respectively? How is the coefficients of the sliding mode controller and the PD controller designed? It is better to give some explanations.

Reply:

Thank you very much for your valuable suggestion.

The sliding mode controller has excellent anti-interference ability. In order to enhance the anti-interference ability of the unicycle robot, the lateral controller is designed as a sliding mode controller. At the same time, it helps to improve the lateral balance ability. The longitudinal controller is used to ensure the longitudinal balance of the unicycle robot while tracking the set speed. The longitudinal control state variable has not only pitch angle, but also speed value. In order to simplify the lateral controller and let the longitudinal balance track the set speed at the same time, the longitudinal controller is designed as a PD controller. The corresponding content has been added in the first paragraph of Section 3 of the revised manuscript to explain the design of the unicycle robot controller.

The initial coefficients of the controller are given according to the estimated parameters in the dynamics of the designed unicycle robot, and then the performance of the unicycle robot is achieved through minor adjustment. The sliding mode controller can enhance the balance ability and anti-interference ability of the unicycle robot by adjusting the coefficient of discontinuous function . The corresponding content has been added in the first paragraph of section 4 of the revised manuscript to explain the design of controller coefficients.

The modified parts have been updated in the Revised Manuscript.

(4) To illustrate the advantage of the dynamic balance controller based on sliding mode controller, the authors are suggested to evaluate the performance of the presented control method in comparison with the existing method by simulation or experiment.

Reply:

Thank you very much for your comments to make this article more rigorous.

In the revised manuscript, the comparison between the designed controller and the PD controller is added to the simulation and experiment. The longitudinal controller of PD controller is the same as the designed controller. The lateral controller is a PD controller composed of roll angle and roll angular velocity. However, in the dynamic balance simulation of unicycle robot under 3N pulse disturbance, PD controller is difficult to keep the balance. Therefore, the 3N pulse interference curve only has the designed controller curves. In the simulation, continuous stochastic interference is added for comparison. After the unicycle robot is balanced for 4s, a stochastic pulse interference within 1-2N is added every 2s. The curves are shown in Fig. 6. Fig. 6(a), 6(b),6(c)and 6(d) are the comparison curves of roll angle, pitch angle, precession angle and speed respectively. In the case of continuous stochastic interference, the pitch angle and speed of PD controller and designed controller have little influence. In the roll angle, the designed controller has less vibration and is more stable than the PD controller in the continuous stochastic interference. In the precession angle, the designed controller has smaller amplitude and faster in convergence speed than PD controller. It can be seen that the designed controller has better anti-interference ability and is more stable than the PD controller in the case of dynamic balance of the unicycle robot in the continuous stochastic interference.

(a)                                      (b)

(c)                                      (d)

Fig. 6 Curves of stochastic pulse interference in simulation.

In dynamic balance experiment, PD controller is added to compare with the designed controller. The PD controller is the same as in the simulation.  The curves are shown in Fig. 7. Fig. 7(a), 7(b),7(c)and 7(d) are the comparison curves of roll angle, pitch angle, precession angle and speed respectively. It can be seen from the figure that both the PD controller and the designed controller can keep the unicycle robot balanced and stable.

(a)                                      (b)

(c)                                      (d)

Fig. 7 Dynamic balance curves.

In the dynamic balance experiment with 0.18Kg mass block placed, the PD controller is difficult to keep the unicycle robot balanced, so the step interference dynamic balance experiment of the PD controller is not shown here.

In the experiment, the designed controller is compared with the PD controller in the dynamic balance experiment when continuous stochastic interference is added. The curves are shown in Fig. 11. Fig. 11(a), 11(b),11(c)and 11(d) are the comparison curves of roll angle, pitch angle, precession angle and speed respectively. It can be seen from the curves that continuous stochastic interference have a greater impact on roll angle and precession angle. However, after the interference, the roll angle and precession angle can still quickly recover to balance. Compared with PD controller, the designed controller has smaller amplitude in roll angle and precession angle. The interference also affects the pitch angle and speed. The pitching angle can still recover the balance state after interference. Although there is a large deviation in speed after interference, it can slowly recover to the tracking speed. Compared with PD controller, the designed controller has smaller oscillation and faster recovery when the pitch angle is subject to continuous stochastic interference and the speed fluctuates less.

(a)                                      (b)

(c)                                      (d)

Fig. 11 Curves of stochastic pulse interference experiment.

Compared with PD controller, the designed controller has stronger anti-interference ability.

The modified parts in Section 4, Section 5, Figs.6, 7 and 11 have been added in the Revised Manuscript.

(5) Why is the interference of the double gyros unicycle robot simulated using a 0.18 kg mass? How is it related to the problem in practical systems? Try to provide some explanations.

Reply:

Thank you very much for your help in pointing out this problem.

In the process of moving outdoors, the unicycle robot is affected by the step interference caused by small gravel falling on it. The mass block is placed at the side of the unicycle robot when it is moving forward dynamically to simulate the anti-interference ability of the unicycle robot in the face of step interference. The 0.18kg mass block is used to simulate the balance ability of the unicycle robot under the influence of large step interference. The corresponding content has been added in the third paragraph of section 5 of the revised manuscript to explain the design of controller coefficients.

The modified parts have been updated in the Revised Manuscript, and all the modified parts have been highlighted in the Revised Manuscript (with changes marked). Sorry again for the confusion caused by us.

(6) Some stochastic values of the control errors obtained from the simulation and experiment should be given to illustrate the high performance of the presented control method.

Reply:

The advantage of the designed sliding mode controller is to enhance the anti-interference ability of the unbalanced unicycle robot. In order to demonstrate the anti-interference ability of the designed controller, simulation and experiment under continuous stochastic interference are added. And by comparing with PD controller, it shows that the designed controller has stronger anti-interference ability and stability.

In the simulation, continuous stochastic interference is added for comparison. After the unicycle robot is balanced for 4s, a stochastic pulse interference within 1-2N is added every 2s. The curves are shown in Fig. 6. Fig. 6(a), 6(b),6(c)and 6(d) are the comparison curves of roll angle, pitch angle, precession angle and speed respectively. In the case of continuous stochastic interference, the pitch angle and speed of PD controller and designed controller have little influence. In the roll angle, the designed controller has less vibration and is more stable than the PD controller in the continuous stochastic interference. In the precession angle, the designed controller has smaller amplitude and faster in convergence speed than PD controller. It can be seen that the designed controller has better anti-interference ability and is more stable than the PD controller in the case of dynamic balance of the unicycle robot in the continuous stochastic interference.

(a)                                      (b)

(c)                                      (d)

Fig. 6 Curves of stochastic pulse interference in simulation.

In the experiment, the designed controller is compared with the PD controller in the dynamic balance experiment when continuous stochastic interference is added. The curves are shown in Fig. 11. Fig. 11(a), 11(b),11(c)and 11(d) are the comparison curves of roll angle, pitch angle, precession angle and speed respectively. It can be seen from the curves that continuous stochastic interference have a greater impact on roll angle and precession angle. However, after the interference, the roll angle and precession angle can still quickly recover to balance. Compared with PD controller, the designed controller has smaller amplitude in roll angle and precession angle. The interference also affects the pitch angle and speed. The pitching angle can still recover the balance state after interference. Although there is a large deviation in speed after interference, it can slowly recover to the tracking speed. Compared with PD controller, the designed controller has smaller oscillation and faster recovery when the pitch angle is subject to continuous stochastic interference and the speed fluctuates less.

(a)                                      (b)

(c)                                      (d)

Fig. 11 Curves of stochastic pulse interference experiment.

The modified parts have been updated in the Revised Manuscript.

(7) The main innovations of the paper should be summarized in Conclusion.

Reply:

Thank you very much for your valuable suggestion.

The contribution of this paper is to design the lateral sliding film controller and the longitudinal speed tracking controller for the double gyros unicycle robot to achieve its dynamic balance ability and dynamic anti-interference ability. The corresponding content has been added in conclusion of the revised manuscript to summarize the innovation of this article.

The modified parts have been updated in the Revised Manuscript.

(8) There are some typo errors in the paper, such as the sentence after equation (10), the first sentence in Section 5.

Reply:

Thank you for your careful review and valuable suggestion to improve the level of this paper.

First, we would like to apologize for the confusion caused by us. In the revised manuscript, the sentences with errors in the article have been corrected as much as possible. In order to reduce errors in the article and increase the readability of the article, this revised manuscript is edited by subject expert editor. The following are the modifications to the sentences in the article:

1 " however, as a mobile robot, its mobility and anti-interference ability are also worth studying." in the introduction is revised to "However, as a mobile robot, its mobility and anti-interference ability are also worth studying.".

2 "The section 4 provides the simulation results." in the introduction is revised to "Section 4 provides the simulation results.".

3 "It is composed of bottom wheel and frame." in the Section 2.2 is revised to "It composes of the bottom wheel and frame.".

4 "Where λ Is a constant parameter. " in the Section 2.2 is revised to "Where λ is a constant parameter. ".

5 "It can be seen that the roll angle of the unicycle robot fluctuates slightly and keeps balance." in the Section 4 is revised to "It can be seen that the roll angle of the unicycle robot fluctuate slightly and keeps balance.".

6 "According to the designed experimental platform of double unicycle robot, the dynamic balance ability and stability of the designed controller are verified. " in the Section 5 is revised to "The dynamic balance ability and stability of the controller to the robot are verified by using the designed experiment platform of the double gyros unicycle robot.".

7 "Figs. 6 (a), 6 (b), 6 (c) and 6 (d) show the roll angle curve, pitch angle curve, precession angle curve and speed curve of the unicycle robot in the dynamic balance experiment. " in Section 5 is revised to "Figs. 6 (a), 6 (b), 6 (c) and 6 (d) show the roll angle curve, the pitch angle curve, the precession angle curve and the speed curve of the unicycle robot in the dynamic balance experiment. ".

8 "Figs. 7 (a), 7 (b), 7 (c) and 7 (d) show the roll angle curve, pitch angle curve, precession angle curve and speed curve of the dynamic interference balance of the unicycle robot, respectively. " in the Section 5 is revised to "Figs. 7 (a), 7 (b), 7 (c) and 7 (d) show the roll angle curve, the pitch angle curve, the precession angle curve and the speed curve of the dynamic interference balance of the unicycle robot, respectively. ".

9 "Figs. 8 (a), 8 (b), 8 (c) and 8 (d) show the roll angle curve, pitch angle curve, precession angle curve and speed curve of the dynamic interference balance of the unicycle robot, respectively." in Section 5 is revised to "Figs. 8 (a), 8 (b), 8 (c) and 8 (d) show the roll angle curve, the pitch angle curve, the precession angle curve and the speed curve of the dynamic interference balance of the unicycle robot, respectively. ".

10 "Figs. 9 (a), 9 (b), 9 (c) and 9 (d) show the roll angle curve, pitch angle curve, precession angle curve and speed curve of the dynamic interference balance of the unicycle robot, respectively. " in the Section 5 is revised to "Figs. 9 (a), 9 (b), 9 (c) and 9 (d) show the roll angle curve, the pitch angle curve, the precession angle curve and the speed curve of the dynamic interference balance of the unicycle robot, respectively. ".

11 "At the same time, due to the relationship between pitch angle and moving speed, the corresponding equation is designed, and the designed speed tracking equation is taken into the longitudinal controller to ensure that the speed can track the designed speed. " in Section 6 is revised to "At the same time, due to the relationship between the pitch angle and moving speed, the corresponding equation is designed, and the designed speed tracking equation is taken into the longitudinal controller to ensure that the speed can track the designed speed. ".

The modified parts have been updated in the Revised Manuscript. The recommendations you proposed are greatly helpful for us to polish our manuscript.  We appreciate your elaborate efforts in reviewing. Thank you very much!

Authors once again sincerely appreciate for your beneficial and helpful review comments. Thank you again for your time on this paper.

Thank your and best regards.

Dr. Hongzhe Jin

Corresponding anthor

Reviewer 2 Report

The paper shows the design of a dynamic balance control for a double gyros unicycle robot, previously developed and modeled by some of the authors whose details are given in reference [14]. The novelty of the proposed control  is the structure of the controller. The overall controller is composed by two decoupled controllers; one, named by the authors as lateral controller, which   is a sliding mode controller to ensure the dynamic balance of the unicycle  robot and increase its anti-interference ability; and the other one named as longitudinal controller, which is a PD controller plus a speed tracker loop to ensure the dynamic speed tracking of the unicycle robot. The rationale behind of the proposal is based on the assumptions that gyros rotations are constants, the influence of yaw angle has not taken into account on the side direction of the unicycle. Also when the unicycle is balanced, the pitch angle \theta and the the bottom wheel speed can be ignored. The output control variables are: a) for the lateral controller,  the precession angular velocity, that takes into account the position and velocity errors of the roll and the precession  angle; and b) for the longitudinal controller, the torque of the bottom wheel, that takes into account the position and velocity  errors of the pitch angle and the speed error of the unicycle.

 The overall controller is validated via numerical simulations and experimental results using the double gyro unicycle robot previously constructed by some of the authors. The experimental tests included,  lateral, middle and longitudinal interferences, and verified the abilities of the proposal.

 Some minor comments.

 1. The proposal only consider the linear dynamics of the unicycle robot; it should be suitable that  be considered the overall nonlinear dynamics, in a future.

2. The control proposal has been done in an heuristic form; it should be nice to include the stability analysis for linear, and nonlinear overall closed-loop system, also in a future.

Author Response

Responses to Review Remarks

Dear reviewer,

First of all, I would like to express my respect for your responsible and careful peer review. Thank you for your valuable comments and allowing a resubmission of our manuscript.

We are uploading (a) our point-by-point response to the comments (below) , (b) an updated manuscript with Revised.

We have revised the paper in our best efforts. However, we are open to revise further should any further suggestion from the reviewer be offered. Thank you for your time and efforts on this paper.

  1. RESPONSE TO THE REVIEWER

The paper shows the design of a dynamic balance control for a double gyros unicycle robot, previously developed and modeled by some of the authors whose details are given in reference [14]. The novelty of the proposed control  is the structure of the controller. The overall controller is composed by two decoupled controllers; one, named by the authors as lateral controller, which   is a sliding mode controller to ensure the dynamic balance of the unicycle  robot and increase its anti-interference ability; and the other one named as longitudinal controller, which is a PD controller plus a speed tracker loop to ensure the dynamic speed tracking of the unicycle robot. The rationale behind of the proposal is based on the assumptions that gyros rotations are constants, the influence of yaw angle has not taken into account on the side direction of the unicycle. Also when the unicycle is balanced, the pitch angle \theta and the the bottom wheel speed can be ignored. The output control variables are: a) for the lateral controller,  the precession angular velocity, that takes into account the position and velocity errors of the roll and the precession  angle; and b) for the longitudinal controller, the torque of the bottom wheel, that takes into account the position and velocity  errors of the pitch angle and the speed error of the unicycle.

 The overall controller is validated via numerical simulations and experimental results using the double gyro unicycle robot previously constructed by some of the authors. The experimental tests included,  lateral, middle and longitudinal interferences, and verified the abilities of the proposal.

Reply:

Thank you very much for your efforts on handling this paper and for whole evaluation of our work.

 Some minor comments.

  1. The proposal only consider the linear dynamics of the unicycle robot; it should be suitable that  be considered the overall nonlinear dynamics, in a future.

Reply:

Thank you for your careful review and valuable suggestion.

In our next research, we mainly consider the nonlinear characteristics of the unicycle robot. Especially, as a complex and highly coupled of the double gyros unicycle robot, the research on its nonlinear dynamics will be of great challenge and value.

Thank you for your valuable suggestions to enrich our future research.

  1. The control proposal has been done in an heuristic form; it should be nice to include the stability analysis for linear, and nonlinear overall closed-loop system, also in a future.

Reply:

Thank you for your valuable comments to let us have more ideas on future research.

As an unstable system, the main difficulty of unicycle robot is to control its balance. Based on its nonlinear dynamic model, the study of its nonlinear control will be of great help to its balance control. In particular, the stability analysis of the designed nonlinear control is important for the stable operation of the unicycle robot. In the future research, we will design the nonlinear controller of the double gyros unicycle robot, and analyze its stability based on the designed controller.

Authors once again sincerely appreciate for your beneficial and helpful review comments. Thank you again for your time on this paper.

Thank your and best regards.

Dr. Hongzhe Jin

Corresponding anthor

Round 2

Reviewer 1 Report

The manuscript has been improved and now warrants publication in Sensors.